# IPEC-J2 rMdr1a, a New Cell Line with Functional Expression of Rat P-glycoprotein Encoded by Rat Mdr1a for Drug Screening Purposes

**DOI:** 10.3390/pharmaceutics12070673

**Published:** 2020-07-17

**Authors:** Lasse Saaby, Josefine Trasborg, Mikkel A. Rasmussen, Bjørn Holst, Birger Brodin

**Affiliations:** 1Bioneer A/S, Kogle Alle 2, DK-2970 Hørsholm, Denmark; mar@bioneer.dk (M.A.R.); bho@bioneer.dk (B.H.); 2Department of Pharmacy, University of Copenhagen, Universitetsparken 2, DK-2100 Copenhagen, Denmark; josefine.trasborg@gmail.com (J.T.); birger.brodin@sund.ku.dk (B.B.)

**Keywords:** P-glycoprotein (ABCB1), P-gp substrates, efflux transport, in vitro model, species differences, drug delivery, drug disposition, drug screening

## Abstract

The efflux pump P-glycoprotein (P-gp) affects drug distribution after absorption in humans and animals. P-gp is encoded by the multidrug resistance gene (MDR1) gene in humans, while rodents (the most common preclinical animal model) express the two isoforms Mdr1a and Mdr1b. Differences in substrate selectivity has also been reported. Our aim was to generate an in vitro cell model with tight barrier properties, expressing functional rat Mdr1a P-gp, as an in vitro tool for investigating species differences. The IPEC-J2 cell line forms extremely tight monolayers and was transfected with a plasmid carrying the rat Mdr1a gene sequence. Expression and P-gp localization at the apical membrane was demonstrated with Western blots and immunocytochemistry. Function of P-gp was shown through digoxin transport experiments in the presence and absence of the P-gp inhibitor zosuquidar. Bidirectional transport experiments across monolayers of the IPEC-J2 rMDR1a cell line and the IPEC-J2 MDR1 cell line, expressing human P-gp, showed comparable magnitude of transport in both the absorptive and efflux direction. We conclude that the newly established IPEC-J2 rMdr1a cell line, in combination with our previously established cell line IPEC-J2 MDR1, has the potential to be a strong in vitro tool to compare P-gp substrate profiles of rat and human P-gp.

## 1. Introduction

During drug development, there is a species-shift in the model systems, which are used to assess the drug candidate compounds. In the early drug development phase, cell-based in vitro models are used to assess pharmacological effects, toxicology and pharmacokinetic parameters before advancing the drug candidate compounds into more advanced animal models. The cell models are often of human origin or humanized animal cells, where human target proteins, e.g., transport proteins, have been transfected into a cell line of animal origin. From the human or humanized cell-based model systems, drug candidate compounds progress to testing in animal models, where rodents like mice and rats are widely used. Finally, successful drug candidates are evaluated in clinical trials in human volunteers. Thus, during a drug development process, the drug candidate compounds are tested in different model systems with different species origin. A well-known potential risk is that data obtained with a human or humanized in vitro model may not be reproducible in a rodent animal model. Similarly, data obtained from an animal experiment in rats or mice may not be possible to reproduce in a human clinical trial. As an example, the expression of the efflux transporter P-glycoprotein is different between humans and rodents. P-glycoprotein (P-gp) belongs to the ATP-binding cassette family of transporters (ABC-transporters), which utilize ATP-hydrolysis to drive the efflux transport of substrate compounds against their concentration gradient [1,2,3]. In humans, P-gp is encoded by the multidrug resistance gene (MDR1), while two isoforms (Mdr1a and Mdr1b) of P-gp are expressed in mice and rats. Together, these two rodent P-gp isoforms seem to provide a similar function as the human P-gp [4]. Using the Basic Local Alignment Search Tool (BLAST, available from https://blast.ncbi.nlm.nih.gov/Blast.cgi), the rat Mdr1a gene product was found to share the highest degree of homology with human P-gp. Rat Mdr1a has a nucleotide sequence homology of 85% with human MDR1, while the nucleotide sequence homology between rat Mdr1b and human Mdr1 is 83%. At the amino acid sequence level, mouse Mdr1a P-gp shares 87% and Mdr1b shares 80% homology to human P-gp, while amino acid sequences for rat Mdr1a and Mdr1b are 80% homologous to human P-gp [5]. Rat Mdr1a is particularly interesting due to the higher homology with human MDR1 and because it is the predominant isoform expressed in rat brain capillaries [6,7]. The substrate profile for murine P-gp transporter proteins seems to be largely overlapping with that of human P-gp: they predominantly recognize hydrophobic and amphiphilic compounds as substrates. Differences in substrate selectivity of human P-gp and murine P-gp have, however, been reported. The antiepileptic drugs phenytoin and levetiracetam showed polarized transport across monolayers of mouse Mdr1a-transfected Lilly Laboratories Cell Porcine Kidney 1 (LLC-PK1) and Madin-Darby canine kidney (MDCK) cell monolayers, while no polarized transport could be observed across cell monolayers overexpressing human P-gp [8]. Among 53 central nervous system (CNS) acting drugs screened for interaction with human P-gp in MDR1-transfected MDCK II cells and for blood–brain barrier penetration through microdialysis experiments, a discrepancy between data from the humanized in vitro model and the murine animal model was found for four compounds (atomoxetine, olanzapine perphenazine and trifluoperazine) [9]. For these CNS drug compounds, a clear interaction with human P-gp in the MDCK II MDR1 cell monolayers (ER > 3) was observed, but at the same time, a considerable CNS penetration was observed with large unbound drug concentrations in CNS relative to the unbound drug concentrations in plasma (K_p,uu_-values > 2). The observed discrepancy between the data from the humanized in vitro model and the murine animal models may be due to species differences in P-gp substrate selectivity. However, both model systems are complex, and the obtained data reflects more than interaction with P-gp. Animal models are the most complex model systems, and in both rats and mice, several efflux transporters such as breast cancer resistance protein (BCRP, ABCG2) and the multidrug resistance-associated proteins (MRPs, ABCCs) will be expressed at the blood-brain barrier (BBB) in addition to P-gp and contribute to overall drug efflux function. Even the simpler in vitro model is more complex than merely reflecting interactions with P-gp. The MDR1-transfected MDCK II cell line is a canine kidney epithelial cell line transfected with the human MDR1 gene to express human P-gp. However, in addition to human P-gp, the MDCK II cell line will also express endogenous canine versions of P-gp and other efflux transporters (e.g., BCRP and MRPs) [10,11,12], which means that the observed efflux transport will not reflect a clear interaction with human P-gp only. As ABC-type efflux transporters are expressed by most cell types, the interference from endogenous transport systems is not unique for the MDCK II cell line but would be relevant for any cell line of animal origin transfected with a human transporter.

The aim of the present study was to generate an in vitro cell model overexpressing functional rat Mdr1a P-gp with minimal interference from endogenous transporters, by transfecting the porcine intestinal epithelial cell line IPEC-J2 with the rat Mdr1a gene and characterizing the resulting cell line. We have previously shown that it is possible to transfect the IPEC-J2 cell line with the human MDR1 gene to obtain an electrically tight cell line overexpressing human P-gp, while the expression of endogenous efflux transporters was low and the contribution from endogenous efflux transport was negligible [13]. In the present study, we transfected the IPEC-J2 cell line with a construct containing the rat Mdr1a gene sequence. Mdr1a mRNA expression was shown with quantitative polymerase chain reaction (qPCR), while expression of P-gp protein and apical localization was confirmed by Western blotting and immunohistochemistry, respectively. Cell monolayers formed by the rMd1a-transfected IPEC-J2 cells retained high transepithelial electrical resistance (TEER) values and low transport of the hydrophilic transport marker mannitol. Functionality of the expressed P-gp was shown through bidirectional transport experiments with the prototypical P-gp substrate digoxin, which showed large efflux ratios indicative of high P-gp functionality. We compared the P-gp activity of cell monolayers of the rMdr1a-transfected IPEC-J2 cell line with the P-gp activity of cell monolayers of the previously published human MDR1-transfected IPEC-J2 cell line [13]. For this comparison, we used olanzapine as a P-gp substrate, for which species differences have been reported, vinblastine as a clinically relevant P-gp substrate and digoxin as a prototypical P-gp substrate. We conclude that the rMdr1a-transfected cell line may prove to be a useful tool to investigate interaction with rat Mdr1a-encoded P-gp, and in combination with a similar cell line, it can be used to screen for species differences in vitro.

## 2. Materials and Methods

### 2.1. Materials

Plasticware such as T-75 culture flasks and Transwell Permeable supports (1.13 cm^2^, 0.4 μm pore size) were purchased from Corning, Fischer Scientific (Slangerup, Denmark). Hank’s balanced salt solution (HBSS) was purchased from Life Technologies (Taastrup, Denmark), Fetal bovine serum (FBS) was purchased from Gibco, Fischer Scientific (Slangerup, Denmark), penicillin and streptomycin were from Bio Whittaker Cambrex (Vallensbaek, Denmark), while 2-[4-(2-hydroxyethyl) piperazin-1-yl] ethanesulfonic acid (HEPES) was from AppliChem GmbH (Darmstadt, Germany). The radio-labelled isotopes, [^3^H]-Digoxin (26.3 Ci/mmol) and [^14^C]-Mannitol (0.057 Ci/mmol), together with Ultima Gold scintillation fluid, were purchased from PerkinElmer (Boston, MA, USA). [^3^H]-Olanzapine (80 Ci/mmol) and [^3^H]-Vinblastine (20 Ci/mmol) were from American Radio-Labelled Chemicals (St. Louis, MO, USA). All other compounds and reagents were purchased from Sigma-Aldrich (Broendby, Denmark), unless otherwise stated.

### 2.2. Transfection and Cell Culture

The IPEC-J2 cell line used as a transfection host in the present study was a gift from Dr. Klaus Bukhave (former Faculty of Life Sciences, University of Copenhagen). The IPEC-J2 cells were cultured in supplemented Dulbecco’s modified eagle’s medium-AQ (10% FBS, 10 μL·mL^−1^ nonessential amino acids (×100), and 100 U·mL^−1^ to 100 μg·mL^−1^ penicillin-streptomycin solution). The rat Mdr1a cDNA, codon-optimized for expression in Sus scrofa, was synthesized by Geneart (Regensburg, Germany) and inserted in the pcDNA3.1 (+) cloning vector under control of a cytomegalovirus (CMV) promoter (see Appendix A) and purchased from Geneart (Regensburg, Germany). The plasmid was amplified in E. coli and the amplified plasmid was isolated and purified using a QIAGEN-Plasmid endofree Midi prep kit, according to the manufacturer’s protocol (Qiagen, Hilden, Germany). Agarose gel electrophoresis and digestion with XhoI restriction enzyme was used to confirm the purity and identity of the amplified plasmid. 0.5 × 10^6^ IPEC-J2 cells were stably transfected with either 5 µg purified rat MDR1-plasmid or 5 µg empty pcDNA 3.1 (+) plasmid (mock control) using a P3 Primary Cell Nucleofector X Kit L in a Amaxa 4D Nucleofector with Program CA167, according to the manufacturer’s protocol (Lonza, Cologne, Germany). As an additional control of transfection efficiency, cells were nucleofected with 1 µg green fluorescent protein (GFP) plasmid. After transfections, the cells were cultured to confluence and subsequently selected for 10 days by addition of 1 mg/mL Geneticin to the culture medium (now defined as passage 1) to kill cells that did not contain the pcDNA 3.1 (+) plasmid, including the GFP control. rMdr1a-transfected cells were further selected for 4 days in the presence of 2 μg/mL puromycin in the culture medium to kill cells that did not express functional P-gp, including the mock control. The rMdr1a-transfected cells were continuously cultured with 2 μg/mL puromycin in culture flasks to maintain the selection pressure. Cell lines were maintained in polycarbonate culture flasks and split once per week. Cells were seeded on permeable Transwell^®^ inserts (1.13 cm^2^ surface area, 0.4 µm pore size) at a density of 4 × 10^4^ cells/insert and cultured for 15–18 days (with culture medium change three times per week) prior to experimentation. Stability of the transfection was estimated through bi-directional transport experiments (see Section 2.7) with digoxin over a large range of passages (see Section 3.1).

### 2.3. RNA Isolation and Real-Time Quantitative Polymerase Chain Reaction

RNA was isolated from IPEC-J2 rMdr1a and IPEC-J2 Mock cells after 15–18 days of culture on permeable inserts. Cells from six permeable inserts obtained from the same passage were lysed and the RNA was isolated using a GeneElute Total RNA Purification kit and an On-Column DNase I Digestion Set according to the manufacturer’s protocol (with the exception that 25 μL of elution A solution was added instead of 50 μL). Reverse transcription was performed using a High-Capacity cDNA Reverse Transcription Kit according to manufacturer’s protocol (AB Applied Biosystems, Foster City, CA, USA) in a DNA Engine Thermal Cycler (PTC-200, MJ Research, Watertown, MA, USA) set to 25 °C for 10 min, 37 °C for 120 min and 85 °C for 5 min. The protocol was deviated by using 0.5 μL Riboloc as an RNase inhibitor and consequently more nuclease-free water for a total volume of 10 μL Master mix per reaction. Real-time quantitative polymerase chain reaction (qPCR) was performed using a FastStart Essential Green DNA Master kit according to the manufacturer’s protocol (Roche Life Science, Hvidovre, Denmark) and a Light Cycler 96 (Roche Life Science, Mannheim, Germany) set to 45 cycles. A deviation from the protocol was that 5 μL of PCR primer was used in the PCR mix instead of 2 μL. Each reaction solution had a cDNA concentration corresponding to 100 ng/μL template RNA and 1 pmol/μL of the relevant primer pairs. mRNA expression levels were quantified relative to β-actin and glyceraldehyde-3-phosphate dehydrogenase (GAPDH) based on the $2^{-\Delta \Delta C_{T}}$ method [14]. The experiments were performed in triplicate for three passages (*n* = 3, *N_total_* = 9). Primer sequences are shown in Table 1 (Invitrogen, Carlsbad, CA, USA).

### 2.4. Immunocytochemistry

IPEC-J2 rMdr1a cells cultured on permeable inserts for 17 days were fixed in 3% paraformaldehyde in phosphate buffered saline (PBS), permeabilized in 0.1% Triton-x100 in PBS and blocked with 2% bovine serum albumin (BSA) in PBS. The filters were cut out of the inserts into smaller pieces, which were incubated overnight at 4 °C in one of the following primary antibodies: rabbit anti-P-gp Ab129450, rabbit anti-ZO1 Ab59720 or rabbit anti-claudin-5 Ab15106 (all from Abcam, Cambridge, UK), diluted 1:100 in 2% BSA in PBS. The filters were then washed in 2% BSA in PBS and incubated in the fluorescent secondary antibody, goat anti-rabbit Alexa 488 AV1008 (Life Technologies, Eugene, OR, USA), diluted 1:200 in 2% BSA in PBS. The secondary antibody solution was added to 1.5 μM propidium iodide (Molecular Probes, Leiden, The Nederlands) for nuclear staining. For visualization of F-actin, one filter was incubated in Alexa fluor 488 phalloidin (Life Technologies, Eugene, OR, USA), diluted 1:40 in 2% BSA in PBS mixed with 1.5 μM propidium iodide. The filters were washed in 2% BSA in PBS, placed onto microscope slides and examined under a confocal laser scanning microscope (LSM 510, Carl Zeiss, Jena, Germany).

### 2.5. Western Blotting

The protein, from IPEC-J2 rMdr1a and IPEC-J2 Mock-transfected cells grown on permeable inserts for 15–18 days, was isolated and analyzed in a Western blot. For both cell lines, six permeable inserts from three passages were pooled and frozen until the day of the experiment. The cells were lysed, and the protein was extracted using a NP-40 buffer containing 8.6% sucrose, 0.03% (ethylenediaminetetraacetic acid) EDTA, 0.04% egtazic acid (EGTA) and 0.24% Tris-HCl in 2% NP-40. The cells were homogenized, incubated on ice and centrifuged (18,000× *g*) for 10 min at 4 °C. The protein concentration of the supernatant was quantified with the Bicinchoninic Acid Protein assay kit according to the manufacturer’s protocol. Protein samples containing 7.8 μg protein per lane were run on a Mini-Protean tris-glycerine extended (TGX) gel according to the manufacturer’s protocol (Bio-rad, Copenhagen, Denmark). The separated proteins were transferred to a 0.2 μm Poly(vinylidene fluoride) (PVDF) membrane using a Trans-Blot Turbo Transfer Pack and System (Bio-rad, Copenhagen, Denmark). The membrane was cut in two between the 75 and the 100 kDa mark and blocked for 1 h in 5% non-fat milk in Tris-buffered Saline with 0.1% Tween (TBST). The membrane piece with the larger protein fragments (above 100 kDa) was incubated in mouse anti-P-gp nb600-1036 (Novun biological, Littleton, CO, USA), diluted 1:300 in 5% milk-TBST, and the membrane piece with smaller protein fragments (below 75 kDa) was incubated in mouse monoclonal anti-β-actin A5441 in 5% milk-TBST at 4 °C overnight. The membranes were washed 3 times for 5 min in TBST and incubated for 1 h in the secondary antibody goat anti-mouse horseradish peroxidase (HRP) 62-6520 (Invitrogen, Carlsbad, CA, USA), diluted 1:4000 in milk-TBST mixed with streptactin-HRP (Precision Plus Strep Tactin-HRP Conjugate, Bio-rad, Hercules, CA, USA), diluted 1:5000 in milk-TBST. Then, the membranes were washed an additional 3 times for 5 min in TBST and incubated in Amersham ECL prime Western Blotting Detection Reagent (GE Healthcare, Little Chalfont, UK). Immediately after, the blots were visualized in the FluorchemQ image system (Protein Simple, San Jose, CA, USA).

### 2.6. Transepithelial Electrical Resistance

The cell monolayers on permeable inserts were allowed to equilibrate to room temperature for 20 min before the transepithelial electrical resistance (TEER) was measured prior to all experiments. The TEER values across the cell monolayers were measured using a chopstick electrode (Millipore Corporation, Bedford, MA, USA) or an Endohm-12 cup electrode (World Precision Instruments Inc., Sarasota, FL, USA) connected to a voltmeter (EVOM^2^, World Precision Instruments Inc., Sarasota, FL, USA). TEER across empty permeable inserts was 8–25 Ω·cm^2^ using the chopstick electrode and 1–68 Ω·cm^2^ using the cup electrode.

### 2.7. Transport Experiments

The cell monolayers were washed and pre-incubated in transport buffer consisting of 10 mM HEPES, 0.05% BSA and 0.038% sodium bicarbonate in HBSS pH 7.4 for 15 min at 37 °C on a shaking table (Unimax 2010, Heidolph, Schwabach, Germany) with a rotation of 75–90 rpm. The buffer was removed, and the transport experiments were initiated by the addition of donor solution containing radio-labelled compound in transport buffer in a concentration of 0.5 or 1 μCi/mL to either the apical or the basolateral chamber. In P-gp inhibition experiments, 0.4 mM Zosuquidar was included in the donor solution as well as the transport buffer. In all transport experiments, the [^3^H]-labelled test compounds were co-administered with [^14^C]-mannitol. Receiver samples were collected after 15, 30, 45, 60, 90 and 120 min from the basolateral (100 µL) or the apical chamber (50 µL), respectively. The volumes of the collected samples were replaced with blank transport buffer (37 °C). All samples were mixed with 2 mL scintillation fluid and their radioactivity was measured in a liquid scintillation counter (Tri-Carb 2910 TR, PerkinElmer, Waltham, MA, USA).

### 2.8. Data Treatment

All transport experiments were conducted across cell monolayers from three independent passages, and for each passage, treatments were made in triplicates (three permeable inserts with cells). Values are presented as means ± standard error of the mean (SEM).

The transepithelial resistance (TEER) values measured across cell monolayers were standardized by multiplication by the cross-sectional area of the permeable inserts to obtain a unit of Ω·cm^2^.

Based on data from transport experiments, the apparent permeability coefficients (P_app_) were calculated using Equation (1):(1)$$P_{\mathrm{app}}\;=\;\frac{J}{C_{0}}\;=\;\frac{Q_{t}}{C_{0}\mathrm{At}}$$
where J denotes the steady-state flux (nmol·cm^−2^·min^−1^), C_0_ is the initial concentration in the donor compartment, A is the cross-sectional area of the permeable inserts (1.13 cm^2^) and Q_t_ represents the accumulated amount of drug in the receiver compartment at time t in minutes (corrected for the dilution arising from replacement of the sampled volume with blank transport buffer). Steady-state flux was calculated as the slope of the straight part of a plot of Q_t_ versus time, to correct for lag-time effects. P_app_ values were calculated using six sampling time points (t = 15, 30, 45, 60, 90 and 120 min). Efflux ratios were calculated as the ratio between the permeability coefficient in the basolateral to apical direction (P_B-A_) and the permeability coefficient in the apical to basolateral direction (P_A-B_), i.e., efflux ratio = P_B-A_/P_A-B_.

Statistical calculations were performed by comparing group means with either a two-tailed Student’s *t*-test or an analysis of variance (ANOVA) followed by a Dunnett’s multiple comparisons test. *p*-values less than 0.05 were considered significant.

## 3. Results

### 3.1. IPEC-J2 Cells Transfected with Rat Mdr1a Exhibited Marked P-glycoprotein Functionality and High Electrical Resistances

The plasmid encoding the rat Mdr1a gene (rMdr1a-pcDNA 3.1(+), see Appendix A) and the empty plasmid (pcDNA 3.1(+)) were amplified and isolated from *E. coli*. IPEC-J2 wild-type (WT) cells were transfected, as described in the Methods Section. The transfected cells were treated with geneticin for 10 days to select for cells which had taken up the rMdr1a plasmid or the empty vector (mock cells). At this point, the IPEC-J2 mock cells were expanded and cryostored (passage 2). The transfected cells were further treated with the cytotoxic aminonucleoside and P-gp substrate, puromycin, in order to select for cells expressing functional P-glycoprotein. Puromycin treatment has previously been shown to be effective for selecting P-gp-expressing cells [13,15]. IPEC-J2 mock cells died within three days in the presence of 2 μg/mL puromycin, while the rMDR1a-transfected cells survived the treatment. The surviving IPEC-J2 rMdr1a cells were expanded with 2 µg/mL puromycin in the growth medium and subsequently cryopreserved (passage 2). Bidirectional transepithelial transport experiments with digoxin combined with TEER measurements were conducted, to investigate P-gp functionality and barrier function of cell monolayers of IPEC-J2 rMdr1a, IPEC-J2 mock and IPEC-J2 WT cells (Figure 1). TEER measurements showed that monolayers of both IPEC-J2 rMdr1a cells and IPEC-J2 mock cells were electrically tight (Figure 1a). However, the TEER values measured across IPEC-J2 mock cell monolayers (4288 ± 5210 Ω·cm^2^) were significantly lower than those of IPEC-J2 wild-type cells, 14,472 ± 635 Ω·cm^2^ (*p* = 0.0002). We have previously observed a similar drop in TEER across cell monolayers of IPEC-J2 cells transfected with the empty pcDNA 3.1(+) plasmid [13]. With measured TEER-values of 13,952 ± 794 Ω·cm^2^, the electrical resistance across IPEC-J2 rMdr1a cell monolayers was not different from those measured across IPEC-J2 WT cells (*p* = 0.9454). The bidirectional transport experiments with digoxin showed marked differences in P-gp function between the different cell monolayers (Figure 1b). The transport of digoxin in the efflux (B–A) direction across monolayers of IPEC-J2 rMdr1a cells was several-fold higher than the corresponding digoxin transport in the absorptive direction (A–B). The apparent efflux ratio for digoxin transport across IPEC-J2 rMdr1a was 42.6, and significantly higher than 1.0 (*p* = 0.0026), which indicates a marked efflux transport of digoxin across IPEC-J2 rMdr1a cells. The A–B and B–A permeabilities of digoxin were more comparable across monolayers of IPEC-J2 WT and IPEC-J2 mock cells with apparent efflux ratios of 2.1 and 1.8, respectively. The efflux ratio for IPEC-J2 mock cells was close to, but less than 2, which has been suggested as a cut-off value for active efflux [16]. The efflux ratio for digoxin transport across IPEC-J2 WT cells was, on the other hand, just above this cut-off value and it would seem that digoxin is actively effluxed by these cells. However, the P_B-A_ value of (1.3 ± 1.7) × 10^−7^ cm s^−1^ for digoxin in IPEC-J2 WT cell monolayers was not significantly different from the corresponding P_A-B_ value of (0.5 ± 0.5) × 10^−7^ cm s^−1^ (*p* = 0.4675). It is therefore unlikely that the observed efflux ratio of 2.1 for digoxin across IPEC-J2 WT cells reflects actual efflux transport.

In another series of bidirectional transport experiments, we demonstrated that the efflux transport of digoxin could be completely blocked by zosuquidar, a third-generation P-gp inhibitor (Figure 2a). The efflux ratio for digoxin was 37.5 in the control condition (transport buffer), while the efflux ratio was reduced to 1.6 in the presence of 0.4 µM zosuquidar (*p* = 0.0019). This finding confirms that the polarized transport of digoxin observed across IPEC-J2 rMdr1a cells was mediated by P-glycoprotein. In similar transport experiments, the presence of 0.4 µM zosuquidar did not affect digoxin transport across monolayers of IPEC-J2 mock cells (Appendix A).

To test the stability of the functional expression of P-gp, IPEC-J2 rMdr1a cells from the same cryopreserved batch were cultured until passage 16; subsequently, bidirectional transport experiments with digoxin as substrate were completed for passages 5–7 and passages 14–16, and the obtained digoxin permeability values and efflux ratios were compared (Figure 2b). The apparent permeability of digoxin in the efflux direction was comparable between the early passages (5–7) and the late passages (14–16) with P_B-A_ values of (1.7 ± 0.1) × 10^−5^ cm s^−1^ and (1.6 ± 0.1) × 10^−5^ cm s^−1^ for the early and late group of passages, respectively. The permeability values in the A–B direction seemed to increase from (3.7 ± 1.0) × 10^−7^ cm s^−1^ in the early passages to (6.8 ± 6.2) × 10^−7^ cm s^−1^. However, the increase was not significant (*p* = 0.0548) in a Student’s *t*-test. The apparent increase in A–B permeability caused the efflux ratio to decrease from 47 in the early passages to 38 in the late passages. However, both efflux ratios are within the range of efflux ratios obtained in previous bidirectional transport experiments with digoxin (see above) and we therefore conclude that the expression of functional P-gp in monolayers of IPEC-J2 Mdr1 cells was stable up to passage 16.

### 3.2. IPEC-J2 rMdr1a Cells Expressed High Levels of Rat Mdr1a and Exhibited Polarized Expression of P-gp in the Apical Membrane

Expression of rat Mdr1a as well as endogenous efflux transporters (MDR1, BCRP and MRPs 1–3) in IPEC-J2 rMdr1a and IPEC-J2 mock cells was investigated by real-time PCR (Figure 3). Real-time PCR data indicated a high expression of mRNA transcript for rat Mdr1a at a level of 13.1% ± 6.9% relative to GAPDH and β-actin in IPEC-J2 rMdr1a cell monolayers, while rat Mdr1a expression could not be detected in IPEC-J2 mock cells (Figure 3a). Expression of porcine MDR1 was not significantly different between IPEC-J2 rMdr1a cells and IPEC-J2 mock cells (Figure 3a, *p* = 0.2461). The apparent expression level of rat Mdr1a in the IPEC-J2 rMdr1a cell monolayers, as indicated by the real-time PCR data, was more than 650-fold higher than the expression of porcine MDR1. The apparent expression level of porcine BCRP was more than 25-fold lower than the expression of rat Mdr1a (Figure 3b). Apparent expression levels of porcine MRP1, 2 and 3 were 33-fold, 4-fold and 6-fold lower than rat Mdr1a expression, respectively (Figure 3b).

The high expression of rat Mdr1a mRNA found in the PCR analysis of IPEC-J2 rMdr1a cells was confirmed by Western blotting (Figure 1c). A distinct band appearing near the 150 kD ladder mark (estimated molecular weight of Mdr1a is 170 kDa) was present in three replicates of IPEC-J2 rMdr1a cell lysates, while no bands could be observed in cell lysates of IPEC-J2 mock cells. β-actin was observed around 40 kDa and was used as a loading control.

The rat Mdr1a-transfected IPEC-J2 cells were further examined by means of immunocytochemistry (Figure 4). Visualization of F-actin revealed that IPEC-J2 rMdr1a cells formed monolayers of flattened cells with a cross-sectional area between 150 and 350 µm^2^ and a cell height of 10–18 µm. The tight junctional proteins ZO-1 and claudin-5 were expressed and both proteins were mainly localized to the cell junctional zones. The localization of ZO-1 appeared to be less defined compared to claudin-5. Staining for P-gp showed expression of P-gp and the xz-stack indicated that P-gp was predominantly expressed in the apical membrane (Figure 4). In this way, the visualization of tight junctional proteins and P-gp confirmed the electrical resistances and the polarized transport of digoxin observed for monolayers of IPEC-J2 rMdr1a cells.

### 3.3. Bidirectional Transport of Selected Model Compounds across Monolayers of IPEC-J2 rMdr1a Cells Was Not Significantly Different from Bidirectional Transport across Monolayers of IPEC-J2 MDR1 Cells Overexpressing Human P-gp

We compared the transport characteristics of IPEC-J2 rMdr1a cell monolayers to that of IPEC-J2 MDR1 cell monolayers overexpressing P-gp through bidirectional transport experiments with four model compounds: digoxin, vinblastine, olanzapine and mannitol (Figure 5). While digoxin, vinblastine and olanzapine are known, or reported, P-gp substrates [10,17,18,19,20,21,22,23], mannitol is known as a low permeable non-substrate compound [13,17,24]. In addition, olanzapine has been shown to be effluxed by MDCK II cells overexpressing human P-gp, while K_p,uu_-values found in rats and mice were higher than unity [9]. The discrepancy between observations of in vitro efflux transport in a cell model overexpressing human P-gp and K_p,uu_-values > 1 in rodent animal models could indicate species differences in the efflux transport of olanzapine. This compound was therefore particularly interesting to include as a model compound in the present study (Figure 5a).

Across IPEC-J2 rMdr1a cell monolayers, the apparent permeability of olanzapine was (4.4 ± 0.3) × 10^−5^ cm s^−1^ in the B–A direction and (2.4 ± 0.3) × 10^−5^ cm s^−1^ in the A–B direction. This corresponded to an efflux ratio of 1.9, indicating that olanzapine was not actively effluxed across monolayers of IPEC-J2 rMdr1a cells. Similar observations were done across IPEC-J2 MDR1 cells, where the apparent permeability of olanzapine was (4.0 ± 0.3) × 10^−5^ cm s^−1^ in the B–A direction and (2.3 ± 0.4) × 10^−5^ cm s^−1^ in the A–B direction, which resulted in an efflux ratio of 1.8. Directly compared, the B–A transport of olanzapine across IPEC-J2 rMdr1a cells was not significantly different from the B–A transport across IPEC-J2 MDR1 cells (*p* = 0.9302), and in a similar manner, the A–B transport across IPEC-J2 rMdr1a cells was not significantly different from the A–B transport across IPEC-J2 MDR1 cells (*p* = 0.7111). Thus, the bidirectional transport experiments with olanzapine across the IPEC-J2 rMdr1a cell monolayers overexpressing rat P-gp and across the IPEC-J2 MDR1 cells overexpressing human P-gp did not indicate species differences.

Digoxin, on the other hand, exhibited clear efflux transport across both cell lines with efflux ratios of 61.4 across IPEC-J2 rMdr1a cell monolayers and 75.7 across IPEC-J2 MDR1 cells (Figure 5b). The A–B and B–A transport of digoxin across IPEC-J2 rMdr1a cells was (3.5 ± 0.3) × 10^−7^ cm s^−1^ and (2.1 ± 0.2) × 10^−5^ cm s^−1^ respectively, while the A–B and B–A transport of digoxin across IPEC-J2 MDR1 cells was (2.8 ± 0.3) × 10^−7^ cm s^−1^ and (2.1 ± 0.2) × 10^−5^ cm s^−1^, respectively. No significant difference in efflux transport of digoxin was observed between the two cell lines (*p* = 0.9614). The bidirectional transport experiments with digoxin across IPEC-J2 rMdr1a and IPEC-J2 MDR1 demonstrated that the observed in vitro efflux transport of a P-gp substrate in cell monolayers overexpressing rat P-gp was not different from efflux transport of the same substrate across monolayers of a similar cell line overexpressing human P-gp.

Vinblastine, a *vinca* alkaloid chemotherapeutic drug compound and a well-known P-gp substrate [17,18,19,20], exhibited marked efflux transport across monolayers of both cell lines (Figure 5c), with efflux ratios of 12.8 across IPEC-J2 rMdr1a cell monolayers and 12.4 across monolayers of IPEC-J2 MDR1 cells. The B–A apparent permeability across IPEC-J2 rMdr1a cells was (2.0 ± 0.5) × 10^−5^ cm s^−1^ and not significantly different from the B–A apparent permeability of (1.9 ± 0.6) × 10^−5^ cm s^−1^ across IPEC-J2 MDR1 cells (*p* = 0.8142). Similarly, the A–B apparent permeability of (1.6 ± 0.2) × 10^−6^ cm s^−1^ across monolayers of IPEC-J2 rMdr1a cells was not significantly different from the corresponding apparent permeability of (1.5 ± 0.2) × 10^−6^ cm s^−1^ across IPEC-J2 MDR1 cells (*p* = 0.8475).

Finally, we tested the bidirectional transport of mannitol, a small hydrophilic compound, which can be used to probe for barrier properties and as a marker for paracellular diffusion (Figure 5d). Mannitol exhibited efflux ratios of 1.5 across monolayers of IPEC-J2 rMdr1a cells and 1.4 across IPEC-J2 MDR1 cell monolayers, which indicated that mannitol was not actively effluxed across the monolayers as the observed efflux ratios were below 2. The A–B apparent permeability was (3.3 ± 1.2) × 10^−7^ cm s^−1^ across IPEC-J2 rMdr1a cell monolayers and not significantly different from the A–B apparent permeability of (2.5 ± 0.2) × 10^−7^ cm s^−1^ across cell monolayers of IPEC-J2 MDR1 cells (*p* = 0.2786). The B–A apparent permeability was (4.4 ± 1.1) × 10^−7^ cm s^−1^ across IPEC-J2 rMdr1a cells and (3.7 ± 0.4) × 10^−7^ cm s^−1^ across IPEC-J2 MDR1 cells, with an insignificant difference between the two (*p* = 0.3845). Overall, the bidirectional data obtained for mannitol across IPEC-J2 rMdr1a and IPEC-J2 MDR1 cells show that mannitol permeated the cell monolayers of both cell lines passively with a low permeability. This observation corroborates the high TEER values measured across the IPEC-J2 rMdr1a cell monolayers in the present study and those reported for the IPEC-J2 MDR1 cells by Saaby et al. [13].

In summary, despite the previous reports of efflux transport across MDCK II cells overexpressing human P-gp, olanzapine was neither a substrate for P-gp across IPEC-J2 rMdr1a cell monolayers nor across IPEC-J2 MDR1 cells overexpressing rat P-gp and human P-gp, respectively. Moreover, the bidirectional transport data observed for olanzapine was not different between the two models, which indicates that the interaction with rat Mdr1a P-gp is not different from human P-gp. This was also observed for the known P-gp substrates digoxin and vinblastine. Both compounds exhibited marked and quantitatively similar efflux transport across monolayers of the two cell lines.

## 4. Discussion

### 4.1. IPEC-J2 rMdr1a Cells Express High Levels of Functional Rat P-glycoprotein

There is a risk that P-gp substrate profiling data produced in humanized in vitro models and clinical trials with human volunteers may not correlate with data from animal studies using rodents such as mice and rats. This can in part be due to differences in P-gp expression levels between humans and experimental animal models and between in vitro and in vivo models but may in part also be due to differences in amino acid sequences between P-glycoproteins expressed in different animal species. As a consequence, there is an interest in assessing the impact of species differences in P-gp function. As an example, cell lines overexpressing the human P-gp and cell lines overexpressing the murine P-gp (encoded by the Mdr1a gene) have been employed to assess species differences in efflux transport profiles for different substrates [8]. Along this line of thought, efflux transport data from cell lines overexpressing human P-gp has been compared to BBB penetration data from murine animal models to study species differences in efflux transporter function. However, these transfected cell lines often express endogenous efflux transporters in addition to expression of the transfected human transporter. Animal models, unless knocked-down, will also express efflux transporters such as BCRP (ABCG2) and MRPs (ABCCs) in addition to, for example, P-gp. Due to a considerable overlap in substrate profiles between the different ABC-type efflux transporters [25,26], the interpretation and comparison of transport data from model systems expressing several efflux transporters can be a difficult task. In the present study, we generated a cell line overexpressing rat P-gp by transfecting the IPEC-J2 cell line with a plasmid containing the rat Mdr1a gene. The high expression of Mdr1a mRNA and the consistently high efflux ratios obtained from bidirectional transport experiments with digoxin strongly suggest that the IPEC-J2 rMdr1a cells expressed high levels of functional P-gp. Bidirectional transport experiments at early and late passages show that the expression of functional P-gp remained stable over several cell passages. The specific P-gp inhibitor zosuquidar abolished vectorial digoxin transport (reduced efflux ratios to unity), which demonstrates that the observed efflux transport of digoxin was mediated solely by P-gp. The high TEER values and low apparent permeability of mannitol across monolayers of IPEC-J2 rMdr1a cell monolayers suggest formation of an extremely tight barrier. TEER values reported for Caco-2 cells and MDCK cells, frequently used cell models to study drug transport, range from 240 to 1200 Ω·cm^2^ and <100 to 10,000 Ω·cm^2^ [10,18,24,27,28,29], respectively. In comparison, the TEER values of 13,952 ± 794 Ω·cm^2^ of IPEC-J2 rMdr1a cell monolayers suggest that this cell line forms a much tighter barrier than the Caco-2 and MDCK MDR1 cell lines. In summary, the bidirectional transport data with digoxin, the expression studies and the measured electrical resistance values demonstrate that the IPEC-J2 rMdr1a cells form tight monolayers with high levels of functional rat P-gp. Overall, the performance of the IPEC-J2 rMdr1a cells, in terms of P-gp functionality and barrier properties, is consistent with the similar cell line IPEC-J2 MDR1, previously established by the group [13].

### 4.2. Expression of rMdr1a Encoding for Rat P-glycoprotein Was Higher than Selected Endogenous Efflux Transporters in IPEC-J2 rMdr1a Cells

The IPEC-J2 rMdr1a cells offer the combination of exceptionally tight monolayers and at the same time, a high expression of functional rat P-gp. Real-time PCR showed that the expression of mRNA for rMdr1a was markedly higher than the expression of endogenous porcine MDR1. This is comparable to our previously established cell line IPEC-J2 MDR1 overexpressing human P-gp, where the expression of hMDR1 mRNA was more than a ~1000-fold higher than the expression of porcine MDR1 mRNA [13]. Expression of rat Mdr1a was 26-fold higher than endogenous porcine BCRP and 33-fold, 4-fold and 6- fold higher than the expression of MRP1, MRP2 and MRP3 mRNA, respectively. Several studies have reported on the use of the LLC-PK1 cell line as a transfection host for rat and mouse Mdr1a. Booth-Genthe and co-workers [30] transfected LL-PK1 cells with rat Mdr1a and found that vincristine-resistance clones exhibited reduced accumulation of the P-gp substrate Rhodamine 123. Based on expression levels and function, one clone was selected, for which P-gp-mediated transport of vinblastine, ritonavir and dexamethasone was demonstrated. However, the researchers did not report on the expression level of rat Mdr1a and endogenous MDR1 or other efflux transporters [30]. Baltes et al. transfected LL-PK1 cells with human MDR1, mouse Mdr1a and mouse Mdr1b and used Western blotting and laser-scanning confocal microscopy to demonstrate expression of P-gp in the transfected cell lines. Reproducible expression of mouse Mdr1a and human MDR1, but not mouse Mdr1b, was observed. Expression of endogenous porcine P-gp was not detectable in LLC-PK1 WT cells [8]. Takeuchi and co-workers transfected LLC-PK1 cells with rat Mdr1a, rat Mdr1b, mouse Mdr1a, mouse Mdr1b, human MDR1, canine MDR1 and monkey MDR1 and used reverse transcriptase PCR and Western blotting to demonstrate expression of P-gp in the respective transfected cells. Expression of endogenous porcine MDR1 mRNA and protein was not demonstrated in either LLC-PK1 WT cells nor in mock-transfected cells. Efflux transport was demonstrated for a range of P-gp substrates, including clarithromycin, daunorubicin, digoxin, erythromycin etoposide, paclitaxel quinidine, ritonavir, saquinavir, verapamil and vinblastine, in all transfected cell lines [31]. Common for the above-referenced studies is that they used non- or semi-quantitative methods to assess the expression of genes encoding for P-gp, which makes it difficult to evaluate the relative expression level between the transfected gene and the endogenous gene. Furthermore, neither of the studies reported on the expression of other endogenous ABC-type efflux transporters. In a study by Kuteykin-Teplyakov et al., LLC-PK1 cells and MDCK II cells were transfected with the human MDR1 gene and the expression level of MDR1 mRNA was compared to endogenous MDR1 [12]. Kuteykin-Teplyakov demonstrated that both cell lines expressed endogenous MDR1. In LLC-PK1 cells, the expression of human MDR1 was 37-fold higher than endogenous porcine MDR1 and the expression of human MDR1 was 19-fold higher than endogenous canine MDR1 in transfected MDCK II cells [12,13].

The IPEC-J2 rMdr1a cell monolayers offer a combination of an expression level of Mdr1a, which was markedly higher than that of endogenous porcine MDR1, while at the same time forming extremely tight barriers. The tight monolayers minimize the paracellular leak of compounds through the cell barrier, which in turn increases the sensitivity for detecting weak substrates. Using the IPEC-J2 MDR1 cells overexpressing human P-gp, we have previously reported the identification of citalopram and atenolol as P-gp substrates, even though these compounds were identified as non-substrates in other in vitro models [13]. Considering the high expression levels of rat Mdr1a compared to porcine MDR1, BCRP and MRP1, the contribution of these endogenous ABC-type efflux transporters on observed efflux transport may thus be negligible. The contribution from porcine MRP2 and MRP3, on the other hand, may not be insignificant. It is also a limitation of the study that GAPDH and ß-actin were used as reference genes, without testing their stability. We cannot completely rule out that transfection might have changed the baseline mRNA levels of GAPDH and ß-actin, thus making it challenging to compare absolute degrees of expression between transfected and non-transfected cells, as well as between cells transfected with different genes. Ideally, the absolute expression levels and stability of the reference genes should be tested in order to make comparisons of absolute degrees of mRNA expression. However, we have, in the present study, demonstrated that the P-gp efflux activity can be completely abolished in the presence of 0.4 µM zosuquidar. Whether or not the observed efflux transport of substrate is mediated solely by rat P-gp is therefore easily tested by comparing the efflux transport with and without zosuquidar inhibition.

### 4.3. Bidirectional Transport of Selected Model Drugs Was Not Different between IPEC-J2 rMdr1a Cells Overexpressing Rat P-gp and IPEC-J2 MDR1 Cells Overexpressing Human P-gp

To investigate whether known P-gp substrates interact differently with rat and human P-gp, we conducted parallel bidirectional transport experiments with olanzapine, vinblastine, digoxin and mannitol across the IPEC-J2 rMdr1a cell monolayers overexpressing rat P-gp and IPEC-J2 MDR1 cell monolayers, which overexpress human P-gp [13]. Olanzapine is classified as a second-generation antipsychotic agent and belongs to the group of thienobenzodiazepines. Summerfield et al. [9] compared in vitro efflux transport of olanzapine to brain penetration data in rats in the form of calculated unbound brain-to-plasma concentration ratios (K_p,uu_). Using an MDCK II MDR1 cell line overexpressing human P-gp, Summerfield et al. observed significant efflux transport of olanzapine with a reported efflux ratio of 4.9. Based on this finding, a K_p,uu_ value less than 1 would be expected. However, from steady-state intravenous brain distribution experiments in rats, Summerfield and co-workers could calculate a K_p,uu_ value of 2.13, which indicates that the concentration of free unbound olanzapine in the brain is approximately 2-fold larger than the free unbound plasma concentration [9]. Due to this discrepancy between in vitro data from a humanized in vitro cell barrier model and in vivo data from rats, we included olanzapine as a model drug. Overall, we could not observe discernable differences between bidirectional transport of olanzapine across monolayers of IPEC-J2 rMdr1a and IPEC-J2 MDR1 cells (Figure 5a). The calculated efflux ratios for olanzapine were 1.9 across IPEC-J2 rMdr1a cells and 1.8 across IPEC-J2 MDR1 cells, indicating that the efflux transport in the B–A direction was only marginally higher than the transport in the A–B direction in both models. Directly compared, the B–A transport of olanzapine was not significantly different between the two models (*p* = 0.9302), and the same was true for the A–B transport (*p* = 0.7111). Based on these findings, we can conclude that the bidirectional transport of olanzapine across IPEC-J2 rMdr1a cells, which overexpressed rat P-gp, was not discernably different from bidirectional transport observed across IPEC-J2 MDR1 cells overexpressing human P-gp. In addition to the rat K_p,uu_ value larger than unity, Summerfield et al. also reported a K_p,uu_ value of 6.9 in mice [9]. K_p,uu_ values larger than unity could indicate active uptake transport of olanzapine across the BBB in the rodent animal models. This is corroborated by a report by Dickens et al., who found that clozapine, a second-generation antipsychotic structurally similar to olanzapine, was actively taken up by the human brain endothelial cell line hCMEC/D3 [32]. It is therefore not unlikely that the discrepancy between in vitro observation of efflux transport of olanzapine and K_p,uu_ values larger than unity in animal models can be explained by active uptake processes outcompeting potential weak efflux transport, rather than species differences. We also conducted parallel bidirectional transport with two known P-gp substrates, digoxin and vinblastine [13,17,18,19,20], across monolayers of the IPEC-J2 rMdr1a and IPEC-J2 MDR1 cell monolayers. Similar to olanzapine, we did not observe significant differences in the bidirectional transport of vinblastine across the two cells overexpressing either rat P-gp or human P-gp. The efflux ratio for vinblastine was 12.8 across IPEC-J2 rMdr1a cell monolayers, while it was 12.4 across monolayers of IPEC-J2 MDR1 cells. Correspondingly, the B–A transport of vinblastine was of similar magnitude across monolayers of the two cell lines (*p* = 0.8142). This is in concordance with findings of Takeuchi and co-workers, who found that the efflux ratio for vinblastine was of similar magnitude across LLC-PK1 cells transfected with rat Mdr1a (18.4) and LLC-PK1 transfected with human MDR1 (12.5) [31]. In the same manner, the observed B–A transport of digoxin across IPEC-J2 rMdr1a cells was not significantly different from the B–A transport observed across IPEC-J2 MDR1 cells (*p* = 0.9614). However, due to slight differences in the A–B transport between the two cell lines, an efflux ratio of 61.4 was obtained across monolayers of Mdr1a cells, while the calculated efflux ratio was 75.7 across MDR1 cell monolayers. In comparison, Takeuchi et al. reported a digoxin efflux ratio of 20.9 across LLC-PK1 cells transfected with rat Mdr1a and 15.3 across LLC-PK1 cells transfected with human MDR1 [31], indicating efflux transport of similar magnitude across the two cell models, which supports our findings for digoxin in the present study. Finally, the bidirectional transport data for mannitol was also of similar magnitude across cell monolayers of both IPEC-J2 rMdr1a and IPEC-J2 MDR1 cells and confirmed that this compound was not a substrate for P-gp (neither rat P-gp nor human P-gp). As an example, the B–A transport of mannitol across IPEC-J2 rMdr1a cell monolayers ((4.4 ± 1.1) × 10^−7^ cm s^−1^) was compared with the B-A transport across cell monolayers of IPEC-J2 MDR1 ((3.7 ± 0.4) × 10^−7^ cm s^−1^), demonstrating no significant difference (*p* = 0.3845) between monolayers of the two cell lines. The low apparent permeability values observed for mannitol (2.5 × 10^−7^ to 4.4 × 10^−7^ cm s^−1^) were consistent with the high TEER-values measured for the two cell lines.

In the present study, we have generated a cell line, IPEC-J2 rMdr1a, which formed very tight monolayers and had a high expression of functional rat P-gp. In addition, we demonstrated consistent bidirectional transport data for olanzapine, digoxin, vinblastine and mannitol across monolayers of the IPEC-J2 rMdr1a cell line and the similar IPEC-J2 MDR1 cell line overexpressing human P-gp. Remarkably, there were no significant differences between A–B and B–A transport of the respective compounds between the two cell lines. Differences in absolute values for, e.g., clearance, permeability coefficients and efflux ratios can normally vary between different cell-based barrier models due to differences in paracellular tightness and expression levels of transport proteins. As an example, the efflux transport of digoxin across LLC-PK1 cells transfected to express P-glycoproteins of different species (human, monkey, canine, rat and mice isoforms) ranged between 9.2 and 15.6 µL min^−1^ mg protein^−1^ in a study by Takeuchi et al. [31]. Similarly, in the same study, the efflux transport of vinblastine across the different P-gp-expressing cell monolayers varied from 5.6 to 11.8 µL min^−1^ mg protein^−1^. The consistent and similar transport data between the IPEC-J2 rMdr1a and IPEC-J2 MDR1 cell lines observed in the present study is a noteworthy finding. Further studies with the IPEC-J2 rMDR1a cell line will address the contribution from endogenous efflux transporters through a series of bidirectional transport experiments with relevant substrates and inhibitors. However, based on the results from the present study, transport experiments using IPEC-J2 rMdr1a and IPEC-J2 MDR1 cell lines appear to represent a unique screening platform, which can be used to compare substrate profiles of rat Mdr1a and human P-gp in order to identify species differences in P-gp-mediated efflux transport.

## 5. Conclusions

By transfecting the porcine epithelial cell line (IPEC-J2) with rat Mdr1a, we have generated a new cell line—the IPEC-J2 rMdr1a. IPEC-J2 rMdr1a cells formed extremely tight monolayers, as judged from the very high transepithelial electrical resistance (~14–15,000 Ω∙cm^2^) and low paracellular permeability of the low-molecular weight marker mannitol (~3 × 10^−7^ cm s^−1^). Through real-time PCR, Western blotting and immunocytochemistry, we have demonstrated that IPEC-J2 rMdr1a cells express high levels of rat P-gp localized to the apical membrane and that the expression of endogenous porcine ABC-type efflux transporters was low in comparison. In addition, we found that the high functional expression of rat P-gp was stable for at least 14 passages. In bidirectional transport experiments, the IPEC-J2 rMdr1a cells exhibited marked efflux transport of the prototypical P-gp substrate digoxin (efflux ratio of 76) and the known P-gp substrate vinblastine (efflux ratio of 13). Thus, we conclude that the newly established IPEC-J2 rMdr1a cell line represents a promising tool to screen drug compounds for interactions with rat P-gp. In combination with similar cell barrier models overexpressing human P-gp (such as the IPEC-J2 MDR1 cells [13]), the IPEC-J2 rMdr1a cell line may potentially be used to screen for species differences between rat and human P-gp.

## Figures and Tables

**Figure 1 pharmaceutics-12-00673-f001:**
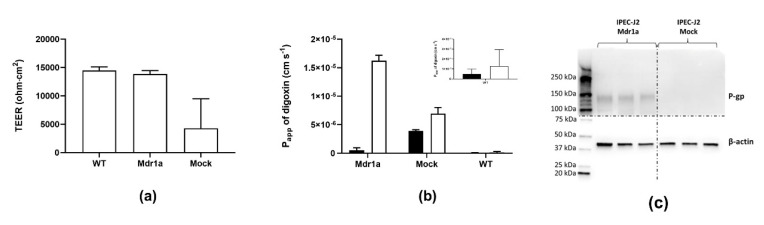
Transendothelial electrical resistance (TEER) measurements, expression of P-glycoprotein (P-gp) in IPEC-J2 rMdr1a monolayers and bidirectional transport of digoxin across IPEC-J2 monolayers. Transport was measured across monolayers formed from IPEC-J2 wild-type cells (WT), mock-transfected IPEC-J2 cells (Mock; transfected with an empty pcDNA vector), and IPEC-J2 cells transfected with rat Mdr1a (Mdr1a). (**a**) TEER was measured across the indicated cell lines after 15–18 days of culture. Measurements were standardized to the area of the permeable support (1.13 cm^2^). Results are shown as means ± SEM of 3–13 individual passages with 6−18 individual permeable supports per passage (*n* = 3–13, N_total_ = 18–174). (**b**) Bidirectional transport of digoxin across the indicated cell lines was measured, and apparent steady-state permeability (P_app_) values were calculated from the steady-state fluxes. The insert shows a magnified scale for digoxin transport across IPEC-J2 WT cells. Filled bars show P_A-B_ values and open bars show P_B-A_ values. Values are given as means ± SEM of 3−6 individual passages with 3 individual permeable supports for each passage and transport direction (*n* = 3–6, N_total_ = 9–18). (**c**) Western blots of protein lysates (7.8 μg) from three different passages (*n* = 3) of IPEC-J2 mock and IPEC-J2 Mdr1a cells. The Western blot was cut in two halves (indicated by the horizontal dotted line), which were developed separately.

**Figure 2 pharmaceutics-12-00673-f002:**
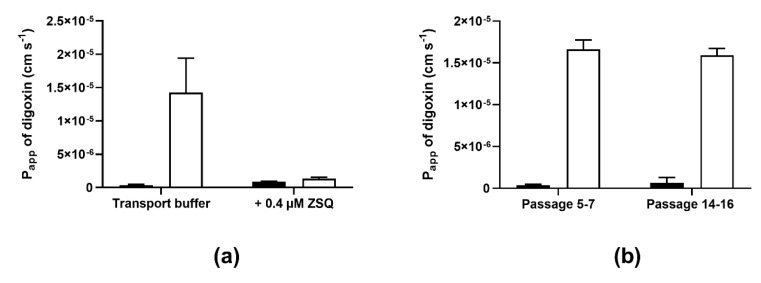
Bidirectional transport of digoxin across IPEC-J2 rat Mdr1a monolayers. (**a**) Bidirectional transport of digoxin across cell monolayers was measured in the presence or absence of the P-gp inhibitor, zosuquidar (ZSQ, 0.4 μM). (**b**) Bidirectional transport of digoxin across IPEC-J2 Mdr1a cells collected from early (5–7) and late (14–16) passages. Apparent permeability (P_app_) values were calculated from the steady-state fluxes, as described in the Methods Section. Filled bars show P_A-B_ values and open bars show P_B-A_ values. Values are given as means ± SEM of 3 individual passages with 3 individual permeable supports for each passage and transport direction (*n* = 3, N_total_ = 9).

**Figure 3 pharmaceutics-12-00673-f003:**
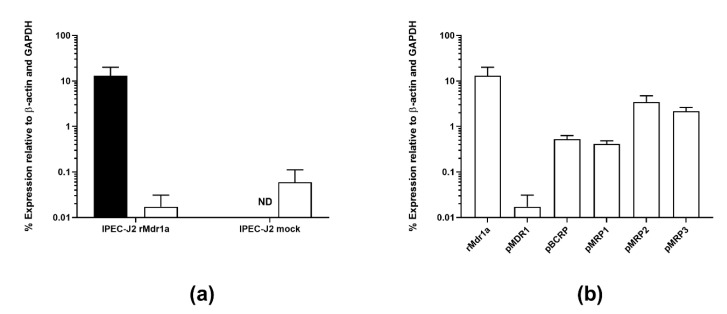
Transporter mRNA expression levels of rat multidrug resistance gene 1a (rMdr1a) and different endogenous porcine efflux transporters in IPEC-J2 cell monolayers. (**a**) Real-time polymerase chain reaction (PCR) data showing mRNA expression levels of rat Mdr1a (rMdr1a) and porcine MDR1 (pMDR1) in IPEC-J2 rMdr1a and mock cell monolayers, quantified relative to β-actin and glyceraldehyde-3-phosphate (GAPDH) expression. Filled bars show expression levels of rat Mdr1a mRNA and open bars show expression levels of porcine MDR1 mRNA. (**b**) Real-time PCR data showing mRNA expression levels of rat Mdr1a (rMdr1a), porcine MDR1 (pMDR1), porcine breast cancer resistance protein (pBCRP), porcine multidrug resistance-associated protein 1 (pMRP1), porcine multidrug resistance-associated protein 2 (pMRP2) and porcine multidrug resistance-associated protein 3 (pMRP3) in IPEC-J2 rMdr1a cells, quantified relative to β-actin and glyceraldehyde-3-phosphate (GAPDH) expression. Bars represent mean values ± SEM from 3 experiments with 3 individual samples per primer-set within each experiment (*n* = 3, total N = 9).

**Figure 4 pharmaceutics-12-00673-f004:**

Localization of filamentous actin (F-actin), claudin-5, zonula occludens-1 (ZO-1) and P-glycoprotein (P-gp) in IPEC-J2 Mdr1a cells. Images show cell monolayers immuno-stained for F-actin, claudin-5, ZO-1 and P-gp (in green) and for propridium iodide (in red) for cell nuclei visibility. Upper row images show xy sections (scale bar = 10 µm), and lower row images show xz sections of the corresponding samples. The lower row images have different scale bars: F-actin = 10 µm, claudin-5 = 2 µm, ZO-1 and P-gp = 5 µm.

**Figure 5 pharmaceutics-12-00673-f005:**
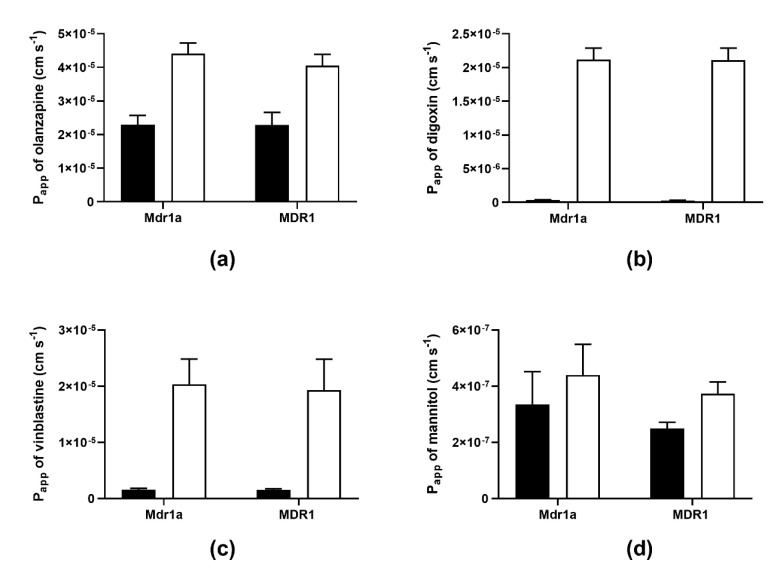
Bidirectional transport of olanzapine (**a**), digoxin (**b**), vinblastine (**c**) and mannitol (**d**) across IPEC-J2 rat Mdr1a (Mdr1a) and IPEC-J2 MDR1 (MDR1) monolayers. Bars represent apparent permeability (P_app_) values calculated from the steady-state fluxes, as described in the Methods Section. Filled bars show P_A-B_ values and open bars show P_B-A_ values. Values are given as means ± SEM of 3–6 individual passages with 3 individual permeable supports for each passage and transport direction (*n* = 3–6, N_total_ = 9–18).

**Table 1 pharmaceutics-12-00673-t001:** Sequence of primers used for quantitative polymerase chain reaction (qPCR).

Protein Name	Target Gene Name	Forward Primer	Reverse Primer	Product Size (bp)
pβ-actin	Atcb	AGGCCAACCGTGAGAAGATG	CATGACAATGCCAGTGGTGC	122
pBCRP	Abcg2	ATCCTGGGCCTGGTTATAGG	GAGACGCTGCTGAAACACTG	170
pClaudin1	CLDN5	TATGACCCCATGACCCCAGT	GGGCCTTGGTGTTGGGTAA	150
pClaudin5	CLDN1	CCCATGTCGCAGAAGTACGA	GGCCGAATACTTGACAGGGA	150
pGAPDH	GAPDH	GTCCACTGGTGTCTTCACGA	TCTCATGGTTCACGCCCATC	125
pMDR1	Abcb1	GCCTCGTATCTTGCTTCTGG	TCAAGTCTGCGTTCTGGATG	146
pMrp-1	Abcc1	AACTTTCTGGCTGGTAGCCC	AGCACGAGGGCGAAGTAAAT	135
pMrp-2	Abcc2	GATGCTCACGTGGGAAGACA	GTGCCATTTCCCACAACCAC	140
pMrp-3	Abcc3	CGTGGCGAGGTGGAGTTC	CATGGATGATTTGCCAGCCC	141
pZO-1	TJP1	CTGAGGGAATTGGGCAGGAA	CCAAAGGACTCAGCAGGGTT	102
rMdr1a	Abcb1a	TTAGATACGGCCGCGAGAAC	TTTCTCCCACGAGTGTGTCG	117

BCRP = breast cancer resistance protein, Mrp = multidrug resistance-associated protein, ZO-1 = Zonula occludens-1, p = porcine target gene, r = rat target gene, bp = base pairs.

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
