# Peer review of "IPEC-J2 rMdr1a, a New Cell Line with Functional Expression of Rat P-glycoprotein Encoded by Rat Mdr1a for Drug Screening Purposes"

_pharmaceutics, 2020, doi:10.3390/pharmaceutics12070673_

Round 1

Reviewer 1 Report

Overall it is an elegant study, and I have only minor comments and suggestions.

Please keep throughout the text Mdr1a – to denominate rat Mdr1a (e.g. change MDR1a-transfected cells – to Mdr1a-transfected, - e.g. line 132, line 137).

Methods – reproducibility of transfection as for P-gp levels, as transient vector – plasmid was used, should be provided.

Results - Figure 1a – How you can explain TEER difference between P-gp expressing and Mock transfected cells.

Discussion – 4.2 chapter title (line 519) “was several fold higher than selected endogenous” does not corroborate with results shown in figure 3b.

Reviewer 2 Report

The authors focused on the preparation of a cell model for monitoring the transport of substances across the cell monolayer by transfecting IPEC-J2 cells with the rat Mdr1a gene. The resulting IPM-J2 rMdr1a cells formed extremely strong monolayers. The overall procedure and description of the experiments is at a standard level and seems plausible. The cells can be used for interesting experiments in the future. However, the description of the preparation of a new cell model brings little new knowledge. Therefore, I lack significant new knowledge in the work of the authors, although I admit that it was performed and described with a high degree of professionalism.

Reviewer 3 Report

Reviewer concerns on this manuscript

Manuscript ID: pharmaceutics-835171

The Manuscript described “IPEC-J2 rMdr1a, A New Cell Line with Functional Expression of Rat P-glycoprotein encoded by rat Mdr1a for Drug Screening Purposes”

The authors were generated an in vitro cell model with tight barrier properties, expressing functional rat Mdr1a P-gp, as an in vitro tool for investigating species differences. The IPEC-J2 cell line forms extremely tight monolayers and was transfected with a plasmid carrying the rat Mdr1a gene sequence. Expression and P-gp localization at the apical membrane was demonstrated with Western blots and immunocytochemistry. Function of P-gp was shown through digoxin transport experiments in the presence and absence of the P-gp inhibitor zosuquidar. Bidirectional transport experiments across monolayers of the IPEC-J2 rMDR1a cell line and the IPEC-J2 MDR1 cell line, expressing human P-gp, showed comparable magnitude of transport in both the efflux and absorptive direction.

The authors also stated, that the newly established IPEC-J2 rMdr1a cell line, in combination with previously established cell line IPEC-J2 MDR1, has the potential to be a strong in vitro tool to compare P-gp substrate profiles of rat and human P-gp.

The design and concept of the article seems pretty good for the current researchers as the authors made an important effort on discovering IPEC-J2 rMDR1a cell line and the IPEC-J2 MDR1 cell line as potential invitro tool.

This investigation almost looks like the article published from the same group a few years ago. (Mol. Pharmaceutics 2016, 13, 2, 640–652)

Nevertheless, there are several major & minor problems with this paper that preclude its publication.

Major concerns:

  1. In Figure 1a. Mock control showed less TEER and needs explanation for this observation. Western blot for confirmation of stable Mrd1a cell line need to be moved into Figure.1 from Figure.4.
  2. As authors performed a single western blot seems too small to be whole figure. The different panels need to be done for the better understand the studies. Also Figure 4. The Western blot bands look not convincing. The experiment needs to be repeated as the Mdr1a protein expression level looks blurry, no band is visible.
  3. Figure 2a. should include control experiments with mock control cells to validate the effect was due to Mdr1a.
  4. Figure 2b needs control experiment with mock control cells to compare effect of early and late passages.
  5. Figure 3. qPCR data is expressed in % relative to β-actin and GAPDH controls. This needs to be explained furthermore technical details and if any specific reasons /or advantage of this way of expression over traditional fold change graphs. Since, cells have high fold expression actin mRNA (at least 4-fold i.e. 2 cq values less) compared to GAPDH. This leads to further variation in normalization.
  6. Line 347 describes rat Mdr1a mRNA levels in percentages with almost 50% deviation. This huge difference and line 350 to 353 describes change in fold levels, while graphs clearly do not represent that change.
  7. qPCR in Figure 3b needs to be done with mock control at least if not WT cells. Since in 3a decreased levels of porcine MDR1 was associated with the presence of rat Mdr1a compared to mock control cells.
  8. In qPCR data describes 650-fold high mRNA in IPEC-J2 rat Mdr1a cells but Western blot do not represent that fold change and bands look not clean. Blot needs to be done with loading high levels of proteins.
  9. Figure 5. also needs to include mock control and WT cells immunocytochemistry. Authors in Figure 1a, showed high TEER in both WT and IPEC-J2 ratMdr1a; and low TEER in mock control. Further they also had shown high polarized transport of digoxin in IPEC-J2 ratMdr1a compared to mock control and IPEC-J2 WT cells. Authors statements from line 397 to 399 for confirmation of high TEER and polarized of transport of digoxin in IPEC-J2 ratMdr1a only valid if they perform similar experiments with mock control and IPEC-J2 WT cells.
  10. The authors showed a high expression of Mdr1a mRNA and the consistently high efflux ratios obtained from bidirectional transport experiments with digoxin, strongly suggest that the IPEC-J2 rMdr1a cells expressed high levels of functional P-gp. However, the authors suggested to show the all protein expression for all the genes they have shown for mRNA expression level for the further confirmation.
  11. The Bidirectional transport of olanzapine (a), digoxin (b), vinblastine (c) and mannitol (d) across IPEC-J2 rat Mdr1a (Mdr1a) and IPEC-J2 MDR1 (MDR1) monolayers. However, a few more compounds need to be investigated to justify the permeability.

Minor concerns:

  1. In Figure 3. (a) “Real-time PCR data showing mRNA expression levels of rat Mdr1a (rMdr1a) and porcine MDR1 (pMDR1) in IPEC-J2 rMdr1a and mock cell monolayers, quantified relative to β- actin and glyceraldehyde-3-phosphate (GAPDH) expression. Filled bars show expression levels of rat Mdr1a

mRNA and open bars show expression levels porcine MDR1 mRNA.”

The authors need to label the figure with respect to the text.

  1. In order to better understand the authors also suggested to create a table for the all values in the bidirectional transport experiments. Includes the chemical structures for the compounds evaluated.
  2. In Western blotting experiments the MDR1 cells expressing p-gp needs to be shown as reference.
  3. In the abstract the word “showed comparable magnitude of transport in both the efflux and absorptive direction” could be changed to absorptive and efflux.
  4. From line 48 to 55 lines need to be reorganized to avoid confusion to common audience by incorporating mouse and choosing Mdr1a over Mdr1b doesn’t seem to be justified well since they both have similar homology with human MDR1.
  5. Mouse Mdr1a and Mdr1b have high homology with human MDR1 over rat. Is there any specific reason to select rat model over Mouse model?
  6. As author mentioned, Puromycin selection concentration is different in methods (line 138,139) and results (line 274,275) and minor issues need to be rectified.
  7. Line 131,132 needs to be reframed to describe the presence of GFP in both Mdr1a and Mock control transfections.
  8. All the methods need to be cited the corresponding references.
  9. Initially, the authors mentioned that “We conclude, that the newly established IPEC-J2 rMdr1a cell line, in combination with our previously established cell line IPEC-J2 MDR1, has the potential to be a strong in vitro tool to compare P-gp substrate profiles of rat and human P-gp.”

However, the comparison with established cell line IPEC-J2 MDR1, was missed in the many experiments. The authors suggested to show the comparison in the application.

“Of note, the design strategy and concept of the work looks considerable and the efforts made to discover novel cell line for drug screening seems appreciable. However, the authors need to carefully rectify all the issues stated above before its publication in this journal”.

Round 2

Reviewer 3 Report

Dear Authors, thank you for addressing all the issues with explanation.

Author Response

We thank the reviewer for the constructive comments and we are happy to have provided the reviewer with explanations to the questions raised.